# GLOBAL OPTIMALITY CONDITIONS FOR DEEP NEURAL NETWORKS

**Chulhee Yun, Suvrit Sra & Ali Jadbabaie**
Massachusetts Institute of Technology
Cambridge, MA 02139, USA
`{chulheey,suvrit,jadbabai}@mit.edu`

## ABSTRACT

We study the error landscape of deep linear and nonlinear neural networks with the squared error loss. Minimizing the loss of a deep linear neural network is a nonconvex problem, and despite recent progress, our understanding of this loss surface is still incomplete. For deep linear networks, we present necessary and sufficient conditions for a critical point of the risk function to be a global minimum. Surprisingly, our conditions provide an efficiently checkable test for global optimality, while such tests are typically intractable in nonconvex optimization. We further extend these results to deep nonlinear neural networks and prove similar sufficient conditions for global optimality, albeit in a more limited function space setting.

## 1 INTRODUCTION

Since the advent of AlexNet (Krizhevsky et al., 2012), deep neural networks have surged in popularity, and have redefined the state-of-the-art across many application areas of machine learning and artificial intelligence, such as computer vision, speech recognition, and natural language processing. However, a concrete theoretical understanding of why deep neural networks work well in practice remains elusive. From the perspective of optimization, a significant barrier is imposed by the nonconvexity of training neural networks. Moreover, it was proved by Blum & Rivest (1988) that training even a 3-node neural network to global optimality is NP-Hard in the worst case, so there is little hope that neural networks have properties that make global optimization tractable.

Despite the difficulties of optimizing weights in neural networks, the empirical successes suggest that the local minima of their loss surfaces could be close to global minima; and several papers have recently appeared in the literature attempting to provide a theoretical justification for the success of these models. For example, by relating neural networks to spherical spin-glass models from statistical physics, Choromanska et al. (2015) provided some empirical evidence that the increase of size of neural networks makes local minima close to global minima.

Another line of results (Yu & Chen, 1995; Soudry & Carmon, 2016; Xie et al., 2016; Nguyen & Hein, 2017) provides conditions under which a critical point of the empirical risk is a global minimum. Such results roughly involve proving that if full rank conditions of certain matrices (as well as some additional technical conditions) are satisfied, derivative of the risk being zero implies loss being zero. However, these results are obtained under restrictive assumptions; for example, Nguyen & Hein (2017) require the width of one of the hidden layers to be as large as the number of training examples. Soudry & Carmon (2016) and Xie et al. (2016) require the product of widths of two adjacent layers to be at least as large as the number of training examples, meaning that the number of parameters in the model must grow rapidly as we have more training data available. Another recent paper (Haeffele & Vidal, 2017) provides a sufficient condition for global optimality when the neural network is composed of subnetworks with identical architectures connected in parallel and a regularizer is designed to control the number of parallel architectures.

Towards obtaining a more precise characterization of the loss-surfaces, a valuable conceptual simplification of deep *nonlinear* networks is deep *linear* neural networks, in which all activation functions are linear and the output of the entire network is a chained product of weight matrices with the input vector. Although at first sight a deep linear model may appear overly simplistic, even its opti-

mization is nonconvex, and only recently theoretical results on this problem have started emerging. Interestingly, already in 1989, Baldi & Hornik (1989) showed that some shallow linear neural networks have no local minima. More recently, Kawaguchi (2016) extended this result to deep linear networks and proved that any local minimum is also global while any other critical point is a saddle point. Subsequently, Lu & Kawaguchi (2017) provided a simpler proof that any local minimum is also global, with fewer assumptions than (Kawaguchi, 2016). Motivated by the success of deep residual networks (He et al., 2016a;b), Hardt & Ma (2017) investigated loss surfaces of deep *linear residual* networks and showed every critical point is a global minimum in a near-identity region; subsequently, Bartlett et al. (2017) extended this result to a nonlinear function space setting.

## 1.1 OUR CONTRIBUTIONS

Inspired by this recent line of work, we study deep linear and nonlinear networks, in settings either similar to or more general than existing work. We summarize our main contributions below.

- We provide both necessary and sufficient conditions for a critical point of the empirical risk to be a global minimum. Specifically, Theorem 2.1 shows that if the hidden layers are wide enough, then a critical point of the risk function is a global minimum *if and only if* the product of all parameter matrices is full-rank. In Theorem 2.2, we consider the case where some hidden layers have smaller width than both the input and output layers, and again provide necessary and sufficient conditions for global optimality. In comparison, Kawaguchi (2016) only proves that every critical point of the risk is either a global minimum or a saddle; it is an "existence" result without any computational implication. In contrast, we present *efficiently checkable* conditions for distinguishing the two different types of critical points; we can even use these conditions while running optimization to test whether the critical points we encounter are saddle points or not, if desired. It is also worth noting that such tests are intractable for general nonconvex optimization (Murty & Kabadi, 1987).

- Under the same assumption as (Hardt & Ma, 2017) on the data distribution, namely, a linear model with Gaussian noise, we can modify Theorem 2.1 to handle the population risk. As a corollary, we not only recover Theorem 2.2 in (Hardt & Ma, 2017), but also extend it to a strictly *larger* set, while removing their assumption that the true underlying linear model has a positive determinant.

- Motivated by (Bartlett et al., 2017), we extend our results on deep linear networks to obtain sufficient conditions for global optimality in deep nonlinear networks, although only via a function space view; these are presented in Theorems 4.1 and 4.2.

## 2 GLOBAL OPTIMALITY CONDITIONS FOR DEEP LINEAR NEURAL NETWORKS

In this section, we describe the problem formulation and notations for deep linear neural networks, state main results (Theorems 2.1 and 2.2), and explain their implication.

## 2.1 PROBLEM FORMULATION AND NOTATION

Suppose we have $m$ input-output pairs, where the inputs are of dimension $d_x$ and outputs of dimension $d_y$. Let $X \in \mathbb{R}^{d_x \times m}$ be the data matrix and $Y \in \mathbb{R}^{d_y \times m}$ be the output matrix. Suppose we have $H$ hidden layers in the network, each having width $d_1, \ldots, d_H$. For notational simplicity we let $d_0 = d_x$ and $d_{H+1} = d_y$. The weights between adjacent layers can be represented as matrices $W_k \in \mathbb{R}^{d_k \times d_{k-1}}$, for $k = 1, \ldots, H+1$, and the output of the network can be written as the product of weight matrices $W_{H+1}, \ldots, W_1$ and data matrix $X$: $W_{H+1} W_H \cdots W_1 X$. We consider minimizing the summation of squared error loss over all data points (i.e. empirical risk),

$$\text{minimize} \quad L(W) := \tfrac{1}{2} \left\| W_{H+1} W_H \cdots W_1 X - Y \right\|_{\mathrm{F}}^2, \tag{1}$$

where $W$ is a shorthand notation for the tuple $(W_1, \ldots, W_{H+1})$.

**Assumptions.** We assume that $d_x \leq m$ and $d_y \leq m$, and that $XX^T$ and $YX^T$ have full ranks. These assumptions are common when we consider supervised learning problems with deep neural networks (e.g. Kawaguchi (2016)). We also assume that the singular values of $YX^T(XX^T)^{-1}X$ are all distinct, which is made for notational simplicity and can be relaxed without too much difficulty.

**Notation.** Given a matrix $A$, let $\sigma_{\max}(A)$ and $\sigma_{\min}(A)$ denote the largest and smallest singular values of A, respectively. Let $\mathrm{row}(A)$, $\mathrm{col}(A)$, $\mathrm{null}(A)$, $\mathrm{rank}(A)$, and $\|A\|_{\mathrm{F}}$ be respectively the row

space, column space, null space, rank, and Frobenius norm of matrix $A$. Given a subspace $V$ of $\mathbb{R}^n$, we denote $V^\perp$ as its orthogonal complement. Given a set $\mathcal{V}$, let $\mathcal{V}^c$ denote the complement of $\mathcal{V}$.

Let us denote $k := \min_{i \in \{0,\dots,H+1\}} d_i$, and define $p \in \operatorname{argmin}_{i \in \{0,\dots,H+1\}} d_i$. That is, $p$ is any layer with the smallest width, and $k = d_p$ is the width of that layer. Here, $p$ might not be unique, but our results hold for any layer $p$ with smallest width. Notice also that the product $W_{H+1} \cdots W_1$ can have rank at most $k$. Let $Y X^T (X X^T)^{-1} X = U \Sigma V^T$ be the singular value decomposition of $Y X^T (X X^T)^{-1} X \in \mathbb{R}^{d_y \times d_x}$. Let $\hat{U} \in \mathbb{R}^{d_y \times k}$ be a matrix consisting of the first $k$ columns of $U$.

## 2.2 Necessary and sufficient conditions for global optimality

We now present two main theorems for deep linear neural networks. The theorems describe two sets, one for the case $k = \min\{d_x, d_y\}$ and the other for $k < \min\{d_x, d_y\}$, inside which every critical point of $L(W)$ is a global minimum. Moreover, the sets have another remarkable property that every critical point outside of these sets is a saddle point. Previous works (Kawaguchi, 2016; Lu & Kawaguchi, 2017) showed that any critical point is either a global minimum or a saddle point, without providing any condition to distinguish between the two; here, we take a step further and partition the domain of $L(W)$ into two sets clearly delineating one set which only contains global minima and the other set with only saddle points.

**Theorem 2.1.** *If $k = \min\{d_x, d_y\}$, define the following set*
$$\mathcal{V}_1 := \{(W_1, \dots, W_{H+1}) : \operatorname{rank}(W_{H+1} \cdots W_1) = k\}.$$
*Then, every critical point of $L(W)$ in $\mathcal{V}_1$ is a global minimum. Moreover, every critical point of $L(W)$ in $\mathcal{V}_1^c$ is a saddle point.*

**Theorem 2.2.** *If $k < \min\{d_x, d_y\}$, define the following set*
$$\mathcal{V}_2 := \left\{(W_1, \dots, W_{H+1}) : \operatorname{rank}(W_{H+1} \cdots W_1) = k, \operatorname{col}(W_{H+1} \cdots W_{p+1}) = \operatorname{col}(\hat{U})\right\}.$$
*Then, every critical point of $L(W)$ in $\mathcal{V}_2$ is a global minimum. Moreover, every critical point of $L(W)$ in $\mathcal{V}_2^c$ is a saddle point.*

Theorems 2.1 and 2.2 provide necessary and sufficient conditions for a critical point of $L(W)$ to be globally optimal. From an algorithmic perspective, they provide easily checkable conditions, which we can use to determine if the critical point the algorithm encountered is a global optimum or not. Given that $L(W)$ is nonconvex, it is interesting to have such efficient tests for global optimality, which is not possible in general (Murty & Kabadi, 1987).

In Hardt & Ma (2017), the authors consider minimizing population risk of linear residual networks:
$$\text{minimize} \quad \tfrac{1}{2} \mathbb{E}_{x,y} \left[ \|(I + W_{H+1}) \cdots (I + W_1)x - y\|_F^2 \right],$$
where $d_x = d_1 = \cdots = d_H = d_y = d$. They assume that $x$ is drawn from a zero-mean distribution with a fixed covariance matrix, and $y = Rx + \xi$ where $\xi$ is iid standard Gaussian noise and $R$ is the true underlying matrix with $\det(R) > 0$. With these assumptions they prove that whenever $\sigma_{\max}(W_i) < 1$ for all $i$, any critical point is a global minimum (Hardt & Ma, 2017, Theorem 2.2).

Under the same assumptions on data distribution, we can slightly modify Theorem 2.1 to derive a population risk counterpart, and in fact notice that the result proved in Hardt & Ma (2017) is a corollary of this modification because having $\sigma_{\max}(W_i) < 1$ for all $i$ is a sufficient condition for $(I + W_{H+1}) \cdots (I + W_1)$ having full rank. Moreover, notice that we can remove the assumption $\det(R) > 0$ which was required by Hardt & Ma (2017). We state this special case as a corollary:

**Corollary 2.3** (Theorem 2.2 of Hardt & Ma (2017)). *Under assumptions on data distribution as described above, any critical point of $\tfrac{1}{2} \mathbb{E}_{x,y} \left[ \|(I + W_{H+1}) \cdots (I + W_1)x - y\|_F^2 \right]$ is a global minimum if $\sigma_{\max}(W_i) < 1$ for all $i$.*

We also note in passing that the classical problem of matrix factorization $\min_{U,V} \|UV^T - Y\|_F^2$ is a special case of deep linear neural networks, so our theorems can also be directly applied.

**Remarks.** The previous result (Kawaguchi, 2016) assumed $d_y \leq d_x$ and showed that: 1) every local minimum is a global minimum, and 2) any other critical point is a saddle point. A subsequent paper by Lu & Kawaguchi (2017) proved 1) without the assumption $d_y \leq d_x$, but as far as we know there is no result showing 2) in the case of $d_y > d_x$. We provide the proof for this case in Lemma B.1. In fact, we propose an alternative proof technique for handling degenerate critical points, which is much simpler than the technique presented by Kawaguchi (2016).

## 3 ANALYSIS OF DEEP LINEAR NETWORKS

In this section, we provide proofs for Theorems 2.1 and 2.2.

### 3.1 SOLUTIONS OF THE RELAXED PROBLEM

We first analyze the globally optimal solution of a "relaxation" of $L(W)$, which turns out to be very useful while proving Theorems 2.1 and 2.2. Consider the relaxed risk function

$$L_0(R) = \frac{1}{2} \|RX - Y\|_{\mathrm{F}}^2 \,,$$

where $R \in \mathbb{R}^{d_y \times d_x}$ and $\mathrm{rank}(R) \leq k$. For any $W$, the product $W_{H+1}W_H \cdots W_1$ has rank at most $k$ and setting $R$ to be this product gives the same loss values: $L_0(W_{H+1}W_H \cdots W_1) = L(W)$. Therefore, $L_0$ is a relaxation of $L$ and

$$\inf_{R:\mathrm{rank}(R) \leq k} L_0(R) \leq \inf_W L(W).$$

This means that if there exists $W$ such that $L(W) = \inf_{R:\mathrm{rank}(R) \leq k} L_0(R)$, then $W$ is a global minimum of the function $L$. This observation is very important in proofs; we will show that inside certain sets, any critical point $W$ of $L(W)$ must satisfy $R^* = W_{H+1} \cdots W_1$, where $R^*$ is a global optimum of $L_0(R)$. This proves that $L(W) = L_0(R^*) = \inf_{R:\mathrm{rank}(R) \leq k} L_0(R)$, thus showing that $W$ is a global minimum of $L$.

By restating this observation as an optimization problem, the solution of problem in (1) is bounded below by the minimum value of the following:

$$\begin{array}{ll} \text{minimize} & \frac{1}{2} \|RX - Y\|_{\mathrm{F}}^2 \\ \text{subject to} & \mathrm{rank}(R) \leq k. \end{array} \tag{2}$$

In case where $k = \min\{d_x, d_y\}$, (2) is actually an unconstrained optimization problem. Note that $L_0$ is a convex function of $R$, so any critical point is a global minimum. By differentiating and setting the derivative to zero, we can easily get the unique globally optimal solution

$$R^* = YX^T(XX^T)^{-1}. \tag{3}$$

In case of $k < \min\{d_x, d_y\}$, the problem becomes non-convex because of the rank constraint, but its exact solution can still be computed easily. We present the solution of this case as a proposition and defer the proof to Appendix C due to its technicalities.

**Proposition 3.1.** *Suppose $k < \min\{d_x, d_y\}$. Then the optimal solution to (2) is*

$$R^* = \hat{U}\hat{U}^T YX^T(XX^T)^{-1}, \tag{4}$$

*which is the orthogonal projection of $YX^T(XX^T)^{-1}$ onto the column space of $\hat{U}$.*

### 3.2 PARTIAL DERIVATIVES OF $L(W)$

By simple matrix calculus, we can calculate the derivatives of $L(W)$ with respect to $W_i$, for $i = 1, \ldots, H + 1$. We present the result as the following lemma, and defer the details to Appendix C.

**Lemma 3.2.** *The partial derivative of $L(W)$ with respect to $W_i$ is given as*

$$\frac{\partial L}{\partial W_i} = W_{i+1}^T \cdots W_{H+1}^T (W_{H+1}W_H \cdots W_1 X - Y) X^T W_1^T \cdots W_{i-1}^T, \tag{5}$$

*for $i = 1, \ldots, H + 1$.*

This result will be used throughout the proof of Theorems 2.1 and 2.2. For clarity in notation, note that when $i = 1$, $W_1^T \cdots W_0^T$ is just an identity matrix in $\mathbb{R}^{d_x \times d_x}$. Similarly, when $i = H + 1$, $W_{H+2}^T \cdots W_{H+1}^T$ is an identity matrix in $\mathbb{R}^{d_y \times d_y}$.

We also state an elementary lemma which proves useful in our proofs, whose proof we defer to Appendix C.

**Lemma 3.3.** *1. For any $A \in \mathbb{R}^{m \times n}$ and $B \in \mathbb{R}^{n \times l}$ where $m \geq n$,*

$$\|AB\|_{\mathrm{F}}^2 \geq \sigma_{\min}^2(A) \|B\|_{\mathrm{F}}^2.$$

*2. For any $A \in \mathbb{R}^{m \times n}$ and $B \in \mathbb{R}^{n \times l}$ where $n \leq l$,*

$$\|AB\|_{\mathrm{F}}^2 \geq \sigma_{\min}^2(B) \|A\|_{\mathrm{F}}^2.$$

### 3.3 PROOF OF THEOREM 2.1

We prove Theorem 2.1, which addresses the case $k = \min\{d_x, d_y\}$. First, recall that the set defined in Theorem 2.1 is

$$\mathcal{V}_1 := \{(W_1, \ldots, W_{H+1}) : \operatorname{rank}(W_{H+1} \cdots W_1) = k\}.$$

As seen in (3), the unique minimum point of $L_0$ has rank $k$. So, no point $W \in \mathcal{V}_1^c$ can be a global minimum of $L$. Therefore, by Kawaguchi (2016, Theorem 2.3.(iii)) and Lemma B.1, any critical point in $\mathcal{V}_1^c$ must be a saddle point.

For the rest of our proof, we need to consider two cases: $d_y \leq d_x$ and $d_x \leq d_y$. If $d_x = d_y$, both cases work. The outline of the proof is as follows: we define a new set $\mathcal{W}_\epsilon$, show that any critical point in the set $\mathcal{W}_\epsilon$ is a global minimum, and then show that every $W \in \mathcal{V}_1$ is also in $\mathcal{W}_\epsilon$ for some $\epsilon > 0$. This proves that any critical point of $L(W)$ in $\mathcal{V}_1$ is also a critical point in $\mathcal{W}_\epsilon$ for some $\epsilon > 0$, hence a global minimum.

The following proposition proves the first step:

**Proposition 3.4.** *Assume that $k = \min\{d_x, d_y\}$. For any $\epsilon > 0$, define the following set:*

$$\mathcal{W}_\epsilon := \begin{cases} \{(W_1, \ldots, W_{H+1}) : \sigma_{\min}(W_{H+1} \cdots W_2) \geq \epsilon\}, & \text{if } d_y \leq d_x, \\ \{(W_1, \ldots, W_{H+1}) : \sigma_{\min}(W_H \cdots W_1) \geq \epsilon\}, & \text{if } d_x \leq d_y. \end{cases}$$

*Then any critical point of $L(W)$ in $\mathcal{W}_\epsilon$ is a global minimum point.*

*Proof.* (If $d_y \leq d_x$) Consider (5) in the case of $i = 1$. We can observe that $W_2^T \cdots W_{H+1}^T \in \mathbb{R}^{d_1 \times d_y}$ and that $d_1 \geq d_y$. Then by Lemma 3.3.1,

$$\left\| \frac{\partial L}{\partial W_1} \right\|_{\mathrm{F}}^2 \geq \sigma_{\min}^2(W_{H+1} \cdots W_2) \left\| (W_{H+1} W_H \cdots W_1 X - Y) X^T \right\|_{\mathrm{F}}^2$$

$$\geq \epsilon^2 \left\| (W_{H+1} W_H \cdots W_1 X - Y) X^T \right\|_{\mathrm{F}}^2.$$

By the above inequality, any critical point in $\mathcal{W}$ satisfies

$$\forall i, \frac{\partial L}{\partial W_i} = 0 \Rightarrow (W_{H+1} W_H \cdots W_1 X - Y) X^T = 0,$$

which means that $W_{H+1} W_H \cdots W_1 = Y X^T (X X^T)^{-1}$. The product is the unique globally optimal solution (3) of the relaxed problem in (2), so $W$ is a global minimum point of $L$.

(If $d_x \leq d_y$) Consider (5) for $i = H + 1$. We can observe that $W_1^T \cdots W_H^T \in \mathbb{R}^{d_x \times d_H}$ and that $d_x \leq d_H$. Then by Lemma 3.3.2,

$$\left\| \frac{\partial L}{\partial W_{H+1}} \right\|_{\mathrm{F}}^2 \geq \epsilon^2 \left\| (W_{H+1} W_H \cdots W_1 X - Y) X^T \right\|_{\mathrm{F}}^2,$$

and the rest of the proof flows in a similar way as the previous case. $\qquad\square$

The next proposition proves the theorem:

**Proposition 3.5.** *For any point $W \in \mathcal{V}_1$, there exists an $\epsilon > 0$ such that $W \in \mathcal{W}_\epsilon$.*

*Proof.* Define a new set $\mathcal{W}$, a "limit" version (as $\epsilon \to 0$) of $\mathcal{W}_\epsilon$, as

$$\mathcal{W} := \begin{cases} \{(W_1, \ldots, W_{H+1}) : \operatorname{rank}(W_{H+1} \cdots W_2) = d_y\}, & \text{if } d_y \leq d_x, \\ \{(W_1, \ldots, W_{H+1}) : \operatorname{rank}(W_H \cdots W_1) = d_x\}, & \text{if } d_x \leq d_y. \end{cases}$$

We show that $\mathcal{V}_1 \subset \mathcal{W}$ by showing that $\mathcal{W}^c \subset \mathcal{V}_1^c$. Consider

$$\mathcal{W}^c = \begin{cases} \{(W_1, \ldots, W_{H+1}) : \operatorname{rank}(W_{H+1} \cdots W_2) < d_y\}, & \text{if } d_y \leq d_x, \\ \{(W_1, \ldots, W_{H+1}) : \operatorname{rank}(W_H \cdots W_1) < d_x\}, & \text{if } d_x \leq d_y. \end{cases}$$

Then any $W \in \mathcal{W}^c$ must have $\operatorname{rank}(W_{H+1} \cdots W_1) < \min\{d_x, d_y\} = k$, so $W \in \mathcal{V}_1^c$. Thus, any $W \in \mathcal{V}_1$ is also in $\mathcal{W}$, so either $\operatorname{rank}(W_{H+1} \cdots W_2) = d_y$ or $\operatorname{rank}(W_H \cdots W_1) = d_x$, depending on the cases. Then, we can set

$$\epsilon = \begin{cases} \sigma_{\min}(W_{H+1} \cdots W_2), & \text{if } d_y \leq d_x, \\ \sigma_{\min}(W_H \cdots W_1), & \text{if } d_x \leq d_y. \end{cases}$$

We always have $\epsilon > 0$ because the matrices are full rank, and we can see that $W \in \mathcal{W}_\epsilon$. $\qquad\square$

### 3.4 PROOF OF THEOREM 2.2

In this section we prove Theorem 2.2, which tackles the case $k < \min\{d_x, d_y\}$. Note that this assumption also implies that $1 \le p \le H$.

As for the proof of Theorem 2.1, define

$$\mathcal{V}_1 := \{(W_1, \ldots, W_{H+1}) : \operatorname{rank}(W_{H+1} \cdots W_1) = k\}.$$

The globally optimal point of the relaxed problem (2) has rank $k$, as seen in (4). Thus, any point outside of $\mathcal{V}_1$ cannot be a global minimum. Then, by Kawaguchi (2016, Theorem 2.3.(iii)) and Lemma B.1, it follows that any critical point in $\mathcal{V}_1^c$ must be a saddle point. The remaining proof considers points in $\mathcal{V}_1$.

For this section, let us introduce some additional notations to ease presentation. Define

$$E := (W_{H+1} \cdots W_1 X - Y)X^T \in \mathbb{R}^{d_y \times d_x},$$
$$A_i := W_{i+1}^T \cdots W_{H+1}^T \in \mathbb{R}^{d_i \times d_y}, \ B_i := W_1^T \cdots W_{i-1}^T \in \mathbb{R}^{d_x \times d_{i-1}}, \quad i = 1, \ldots, H+1,$$

so that $\frac{\partial L}{\partial W_i} = A_i E B_i$. Notice that $A_{H+1}$ and $B_1$ are identity matrices.

Now consider any tuple $W \in \mathcal{V}_1$. Since the full product $W_{H+1} \cdots W_1$ has rank $k$, any partial products $A_i$ and $B_i$ must have $\operatorname{rank}(A_i) \ge k$ and $\operatorname{rank}(B_i) \ge k$, for all $i$. Then, consider $A_p \in \mathbb{R}^{k \times d_y}$ and $B_{p+1} \in \mathbb{R}^{d_x \times k}$. Since $\operatorname{rank}(A_p) \le k$ and $\operatorname{rank}(B_{p+1}) \le k$, we can see that $\operatorname{rank}(A_p) = \operatorname{rank}(B_{p+1}) = k$. Also, notice that $A_i = W_{i+1} A_{i+1}$ and $B_{i+1} = B_i W_i$, so that

$$\operatorname{rank}(A_1) \le \operatorname{rank}(A_2) \le \cdots \le \operatorname{rank}(A_p) \text{ and } \operatorname{rank}(B_{H+1}) \le \operatorname{rank}(B_H) \le \cdots \le \operatorname{rank}(B_{p+1}).$$

However, we have $k \le \operatorname{rank}(A_1)$ and $k \le \operatorname{rank}(B_{H+1})$, so the ranks are all identically $k$. Also,

$$\operatorname{row}(A_1) \subset \operatorname{row}(A_2) \subset \cdots \subset \operatorname{row}(A_p) \text{ and } \operatorname{col}(B_{H+1}) \subset \operatorname{col}(B_H) \subset \cdots \subset \operatorname{col}(B_{p+1}),$$

but it was just shown that the these spaces have the same dimensions, which equals $k$, meaning

$$\operatorname{row}(A_1) = \operatorname{row}(A_2) = \cdots = \operatorname{row}(A_p) \text{ and } \operatorname{col}(B_{H+1}) = \operatorname{col}(B_H) = \cdots = \operatorname{col}(B_{p+1}).$$

Using this observation, we can now state a proposition showing necessary and sufficient conditions for a tuple $W \in \mathcal{V}_1$ to be a critical point of $L(W)$.

**Proposition 3.6.** *A tuple $W \in \mathcal{V}_1$ is a critical point of $L$ if and only if $A_p E = 0$ and $E B_{p+1} = 0$.*

*Proof.* (If part) $A_p E = 0$ implies that $\operatorname{col}(E) \subset \operatorname{row}(A_p)^\perp = \cdots = \operatorname{row}(A_1)^\perp$, so $\frac{\partial L}{\partial W_i} = A_i E B_i = 0 \cdot B_i = 0$, for $i = 1, \ldots, p$. Similarly, $E B_{p+1} = 0$ implies $\operatorname{row}(E) \subset \operatorname{col}(B_{p+1})^\perp = \cdots = \operatorname{col}(B_{H+1})^\perp$, so $\frac{\partial L}{\partial W_i} = A_i E B_i = A_i \cdot 0 = 0$ for $i = p+1, \ldots, H+1$.

(Only if part) We have $\frac{\partial L}{\partial W_i} = A_i E B_i = 0$ for all $i$. This means that

$$\operatorname{col}(E B_i) \subset \operatorname{row}(A_i)^\perp = \operatorname{row}(A_p)^\perp \text{ for } i = 1, \ldots, p$$
$$\operatorname{row}(A_i E) \subset \operatorname{col}(B_i)^\perp = \operatorname{col}(B_{p+1})^\perp \text{ for } i = p+1, \ldots, H+1.$$

Now recall that $B_1$ and $A_{H+1}$ are identity matrices, so $\operatorname{col}(E) \subset \operatorname{row}(A_p)^\perp$ and $\operatorname{row}(E) \subset \operatorname{col}(B_{p+1})^\perp$, which proves $A_p E = 0$ and $E B_{p+1} = 0$. $\square$

Now we present a proposition that specifies the necessary and sufficient condition in which a critical point of $L(W)$ in $\mathcal{V}_1$ is a global minimum. Recall that when we take the SVD $Y X^T (X X^T)^{-1} X = U \Sigma V^T$, $\hat{U} \in \mathbb{R}^{d_y \times k}$ is defined to be a matrix consisting of the first $k$ columns of $U$.

**Proposition 3.7.** *A critical point $W \in \mathcal{V}_1$ of $L(W)$ is a global minimum point if and only if $\operatorname{col}(W_{H+1} \cdots W_{p+1}) = \operatorname{row}(A_p) = \operatorname{col}(\hat{U})$.*

*Proof.* Since $W$ is a critical point, by Proposition 3.6 we have $A_p E = 0$. Also note from the definitions of $A_i$'s and $B_i$'s that $W_{H+1} \cdots W_1 = A_p^T B_{p+1}^T$, so

$$A_p E = A_p (A_p^T B_{p+1}^T X - Y)X^T = A_p A_p^T B_{p+1}^T X X^T - A_p Y X^T = 0.$$

Because $\text{rank}(A_p) = k$, and $A_p A_p^T \in \mathbb{R}^{k \times k}$ is invertible, so $B_{p+1}$ is determined uniquely as

$$B_{p+1}^T = (A_p A_p^T)^{-1} A_p Y X^T (XX^T)^{-1},$$

thus

$$W_{H+1} \cdots W_1 = A_p^T B_{p+1}^T = A_p^T (A_p A_p^T)^{-1} A_p Y X^T (XX^T)^{-1}.$$

Comparing this with (4), $W$ is a global minimum solution if and only if

$$\hat{U}\hat{U}^T Y X^T (XX^T)^{-1} = W_{H+1} \cdots W_1 = A_p^T (A_p A_p^T)^{-1} A_p Y X^T (XX^T)^{-1}.$$

This equation holds if and only if $A_p^T (A_p A_p^T)^{-1} A_p = \hat{U}\hat{U}^T$, meaning that they are projecting $Y X^T (XX^T)^{-1}$ onto the same subspace. The projection matrix $A_p^T (A_p A_p^T)^{-1} A_p$ is onto $\text{row}(A_p)$, while $\hat{U}\hat{U}^T$ is onto $\text{col}(\hat{U})$. From this, we conclude that $W$ is a global minimum point if and only if $\text{row}(A_p) = \text{col}(\hat{U})$. $\qquad\square$

From Proposition 3.7, we can define the set $\mathcal{V}_2$ that appeared in Theorem 2.2, and conclude that every critical point of $L(W)$ in $\mathcal{V}_2$ is a global minimum, and any other critical points are saddle points.

## 4 EXTENSION TO DEEP NONLINEAR NEURAL NETWORKS

In this section, we present some sufficient conditions for global optimality for deep nonlinear neural networks via a function space view. Given a smooth nonlinear function $h^*$ that maps input to output, Bartlett et al. (2017) described a method to decompose it into a number of smooth nonlinear functions $h^* = h_{H+1} \circ \cdots \circ h_1$ where $h_i$'s are close to identity. Using Fréchet derivatives of the population risk with respect to each function $h_i$, they showed that when all $h_i$'s are close to identity, any critical point of the population risk is a global minimum. One can see that these results are direct generalization of Theorems 2.1 and 2.2 of Hardt & Ma (2017) to nonlinear networks and utilize the classical "small gain" arguments often used in nonlinear analysis and control (Khalil, 1996; Zames, 1966). Motivated by this result, we extended Theorem 2.1 to deep nonlinear neural networks and obtained sufficient conditions for global optimality in function space.

### 4.1 PROBLEM FORMULATION AND NOTATION

Suppose the data $X \in \mathbb{R}^{d_x}$ and its corresponding label $Y \in \mathbb{R}^{d_y}$ are drawn from some distribution. Notice that in this section, $X$ and $Y$ are random vectors instead of matrices. We want to predict $Y$ given $X$ with a deep nonlinear neural network that has $H$ hidden layers. We express each layer of the network as functions $h_i : \mathbb{R}^{d_{i-1}} \to \mathbb{R}^{d_i}$, so the entire network can be expressed as a composition of functions: $h_{H+1} \circ h_H \circ \cdots \circ h_1$. Our goal is to obtain functions $h_1, \ldots, h_{H+1}$ that minimize the population risk functional:

$$L(h) = L(h_1, \ldots, h_{H+1}) := \frac{1}{2}\mathbb{E}\left[\|h_{H+1} \circ \cdots \circ h_1(X) - Y\|_2^2\right],$$

where $h$ is a shorthand notation for $(h_1, \ldots, h_{H+1})$. It is well-known that the minimizer of squared error risk is the conditional expectation of $Y$ given $X$, which we will denote $h^*(x) = \mathbb{E}[Y \mid X = x]$. With this, we can separate the risk functional into two terms

$$L(h) = \frac{1}{2}\mathbb{E}\left[\|h_{H+1} \circ \cdots \circ h_1(X) - h^*(X)\|_2^2\right] + C,$$

where the constant $C$ denotes the variance that is independent of $h_1, \ldots, h_{H+1}$. Note that if $h_{H+1} \circ \cdots \circ h_1 = h^*$ almost surely, the first term in $L(h)$ vanishes and the optimal value $L^*$ of $L(h)$ is $C$.

**Assumptions.** Define the function spaces as the following:

$$\mathcal{F} := \left\{ h : \mathbb{R}^{d_x} \to \mathbb{R}^{d_y} \mid h \text{ is differentiable, } h(0) = 0, \text{ and } \sup_x \frac{\|h(x)\|_2}{\|x\|_2} < \infty \right\},$$

$$\mathcal{F}_i := \left\{ h : \mathbb{R}^{d_{i-1}} \to \mathbb{R}^{d_i} \mid h \text{ is differentiable, } h(0) = 0, \text{ and } \sup_x \frac{\|h(x)\|_2}{\|x\|_2} < \infty \right\},$$

where $\mathcal{F}_i$ are defined for all $i = 1, \ldots, H + 1$. Assume that $h^* \in \mathcal{F}$, and that we are optimizing $L(h)$ with $h_1 \in \mathcal{F}_1, \ldots, h_{H+1} \in \mathcal{F}_{H+1}$. In other words, the functions in $\mathcal{F}, \mathcal{F}_1, \ldots, \mathcal{F}_{H+1}$ are differentiable and show sublinear growth starting from 0. Notice that $h_{H+1} \circ \cdots \circ h_1 \in \mathcal{F}$, because a composition of differentiable functions is also differentiable, and a composition of sublinear functions is also sublinear. We also assume that $d_i \geq \min\{d_x, d_y\}$ for all $i = 1, \ldots, H + 1$, which is identical to the assumption $k = \min\{d_x, d_y\}$ in Theorem 2.1.

**Notation.** To simplify multiple composition of functions, we denote $h_{i:j} = h_i \circ h_{i-1} \circ \cdots \circ h_{j+1} \circ h_j$. As in the matrix case, $h_{0:1}$ and $h_{H+1:H+2}$ mean identity maps in $\mathbb{R}^{d_x}$ and $\mathbb{R}^{d_y}$, respectively. Given a function $f$, let $J[f](x)$ be the Jacobian matrix of function $f$ evaluated at $x$. Let $D_{h_i}[L(h)]$ be the Fréchet derivative of $L(h)$ with respect to $h_i$ evaluated at $h$. The Fréchet derivative $D_{h_i}[L(h)]$ is a linear functional that maps a function (direction) $\eta \in \mathcal{F}_i$ to a real number (directional derivative).

## 4.2 SUFFICIENT CONDITIONS FOR GLOBAL OPTIMALITY

Here, we present two theorems which give sufficient conditions for a critical point ($D_{h_i}[L(h)] = 0$ for all $i$) in the function space to be a global optimum. The proofs are deferred to Appendix A.

**Theorem 4.1.** *Consider the case $d_x \geq d_y$. If there exists $\epsilon > 0$ such that*

1. *$J[h_{H+1:2}](z) \in \mathbb{R}^{d_y \times d_1}$ has $\sigma_{\min}(J[h_{H+1:2}](z)) \geq \epsilon$ for all $z \in \mathbb{R}^{d_1}$,*

2. *$h_{H+1:2}(z)$ is twice-differentiable,*

*then any critical point of $L(h)$, in terms of $D_{h_1}[L(h)], \ldots, D_{h_{H+1}}[L(h)]$, is a global minimum.*

**Theorem 4.2.** *Consider the case $d_x \leq d_y$. Assume that there exists some $j \in \{1, \ldots, H + 1\}$ such that $d_x = d_{j-1}$ and $d_y \leq d_j$. If there exist $\epsilon_1, \epsilon_2 > 0$ such that*

1. *$h_{j-1:1} : \mathbb{R}^{d_x} \to \mathbb{R}^{d_{j-1}} = \mathbb{R}^{d_x}$ is invertible,*

2. *$h_{j-1:1}$ satisfies $\|h_{j-1:1}(u)\|_2 \geq \epsilon_1 \|u\|_2$ for all $u \in \mathbb{R}^{d_x}$,*

3. *$J[h_{H+1:j+1}](z) \in \mathbb{R}^{d_y \times d_j}$ has $\sigma_{\min}(J[h_{H+1:j+1}](z)) \geq \epsilon_2$ for all $z \in \mathbb{R}^{d_j}$,*

4. *$h_{H+1:j+1}(z)$ is twice-differentiable,*

*then any critical point of $L(h)$, in terms of $D_{h_1}[L(h)], \ldots, D_{h_{H+1}}[L(h)]$, is a global minimum.*

Note that these theorems give *sufficient* conditions, whereas Theorems 2.1 and 2.2 provide *necessary and sufficient* conditions. So, if the sets we are describing in Theorems 4.1 and 4.2 do not contain any critical point, the claims would be vacuous. We ensure that there are critical points in the sets, by presenting the following proposition, whose proof is also deferred to Appendix A.

**Proposition 4.3.** *For each of Theorems 4.1 and 4.2, there exists at least one global minimum solution of $L(h)$ satisfying the conditions of the theorem.*

**Discussion and Future work.** Theorems 4.1 and 4.2 state that in certain sets of $(h_1, \ldots, h_{H+1})$, any critical point in function space a global minimum. However, this does not imply that any critical point for a fixed sigmoid or arctan network is a global minimum. As noted in (Bartlett et al., 2017), there is a downhill direction in function space at any suboptimal point, but this direction might be orthogonal to the function space represented by a fixed network, and may hence result in local minima in the parameter space of the fixed architecture.

Understanding the connection between the function space and parameter space of commonly used architectures is an open direction for future research, and we believe that these results can be good initial steps from the theoretical point of view. For example, we can see that one of the sufficient conditions for global optimality is the Jacobian matrix being full rank. Given that a nonlinear function can locally be linearly approximated using Jacobians, this connection is already interesting. An extension of the function space viewpoint to cover different architectures or design new architectures (that have "better" properties when viewed via the function space view) should also be possible and worth studying.

ACKNOWLEDGMENTS

This research project was supported in parts by DARPA DSO's Fundamental Limits of Learning program.

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

## A   ANALYSIS OF DEEP NONLINEAR NETWORKS

### A.1   NOTATION

In this section, we introduce additional notation that is used in the proofs. To emphasize that the Fréchet derivative $D_{h_i}[L(h)]$ is a linear functional that outputs a real number, we will write $D_{h_i}[L(h)](\eta)$ in an inner-product form $\langle D_{h_i}[L(h)], \eta \rangle$. This notation also helps avoiding confusion coming from multiple parentheses and square brackets.

There are many different kinds of norms that appear in the proofs. Given a finite-dimensional real vector $v$, $\|v\|_2$ denotes its $\ell_2$ norm. For a matrix $A$, its operator norm is defined as $\|A\|_{\mathrm{op}} = \sup_x \frac{\|Ax\|_2}{\|x\|_2}$. Let $h \in \mathcal{F}$. Then define a "generalized" induced norm for nonlinear functions with sublinear growth: $\|h\|_{\mathrm{nl}} = \sup_x \frac{\|h(x)\|_2}{\|x\|_2}$, where the subscript nl is used to emphasize that this norm is for nonlinear functions. The norm $\|\cdot\|_{\mathrm{nl}}$ is defined in the same way for $\mathcal{F}_i$'s. Now, given a linear functional $G$ that maps a function $f \in \mathcal{F}_i$ to a real number $\langle G, f \rangle$, define the operator norm $\|G\|_{\mathrm{op}} = \sup_{f \in \mathcal{F}_i} \frac{\langle G, f \rangle}{\|f\|_{\mathrm{nl}}}$.

### A.2   FRÉCHET DERIVATIVES

By definition of Fréchet derivatives, we have

$$\langle D_{h_i}[L(h)], \eta \rangle = \lim_{\epsilon \to 0} \frac{L(h_1, \ldots, h_i + \epsilon\eta, \ldots, h_{H+1}) - L(h)}{\epsilon},$$

where $\eta \in \mathcal{F}_i$ is the direction of perturbation and $\langle D_{h_i}[L(h)], \eta \rangle$ is the directional derivative along that direction $\eta$. From the definition of $L(h)$,

$$
\begin{aligned}
&L(h_1, \ldots, h_i + \epsilon\eta, \ldots, h_{H+1}) \\
=& \frac{1}{2}\mathbb{E}\left[\|h_{H+1:i+1} \circ (h_i + \epsilon\eta) \circ h_{i-1:1}(X) - h^*(X)\|_2^2\right] + C \\
=& \frac{1}{2}\mathbb{E}\left[\|h_{H+1:i+1}(h_{i:1}(X) + \epsilon\eta(h_{i-1:1}(X))) - h^*(X)\|_2^2\right] + C \\
=& \frac{1}{2}\mathbb{E}\left[\|h_{H+1:1}(X) + \epsilon J[h_{H+1:i+1}](h_{i:1}(X))\eta(h_{i-1:1}(X)) + O(\epsilon^2) - h^*(X)\|_2^2\right] + C \\
=& L(h) + \epsilon\mathbb{E}\left[(h_{H+1:1}(X) - h^*(X))^T J[h_{H+1:i+1}](h_{i:1}(X))\eta(h_{i-1:1}(X))\right] + O(\epsilon^2).
\end{aligned}
$$

Therefore,

$$\langle D_{h_i}[L(h)], \eta \rangle = \mathbb{E}\left[(h_{H+1:1}(X) - h^*(X))^T J[h_{H+1:i+1}](h_{i:1}(X))\eta(h_{i-1:1}(X))\right]. \tag{6}$$

This equation (6) will be used in the proof of Theorems 4.1 and 4.2.

### A.3   PROOF OF THEOREM 4.1

From (6), consider $D_{h_1}[L(h)]$. For any $\eta \in \mathcal{F}_1$,

$$\langle D_{h_1}[L(h)], \eta \rangle = \mathbb{E}\left[(h_{H+1:1}(X) - h^*(X))^T J[h_{H+1:2}](h_1(X))\eta(X)\right].$$

Let $A(X) = J[h_{H+1:2}](h_1(X))$. Since $A(X)$ has full row rank by assumption, $A(X)A(X)^T$ is invertible. Then define a particular direction

$$\tilde{\eta}(X) = A(X)^T(A(X)A(X)^T)^{-1}(h_{H+1:1}(X) - h^*(X)),$$

so that

$$\langle D_{h_1}[L(h)], \tilde{\eta} \rangle = \mathbb{E}\left[\|h_{H+1:1}(X) - h^*(X)\|_2^2\right].$$

It remains to check if $\tilde{\eta} \in \mathcal{F}_1$. It is easily checked that $\tilde{\eta}(0) = 0$ because $h_{H+1:1}(0) - h^*(0) = 0$. Since $J[h_{H+1:2}]$ is differentiable by assumption and $h_1 \in \mathcal{F}_1$, $A(X)$ is differentiable and $A(X)^T$, $(A(X)A(X)^T)^{-1}$ are differentiable functions. Also, $h_{H+1:1} - h^* \in \mathcal{F}$, so we can conclude that $\tilde{\eta}$ is differentiable.

Moreover, if we decompose $A(X)$ with SVD, $A(X) = U\Sigma V^T$, $\Sigma$ is of the form $\Sigma = [\Sigma_1 \ 0]$ and

$$A(X)^T (A(X)A(X)^T)^{-1} = V\Sigma^T U^T (U\Sigma V^T V\Sigma^T U^T)^{-1} = V\Sigma^T U^T (U\Sigma_1^2 U^T)^{-1}$$
$$= V\Sigma^T U^T U\Sigma_1^{-2} U^T = V \begin{bmatrix} \Sigma_1^{-1} \\ 0 \end{bmatrix} U^T,$$

from which we can see that

$$\left\| A(X)^T (A(X)A(X)^T)^{-1} \right\|_{\mathrm{op}} = \sigma_{\max}(A(X)^T (A(X)A(X)^T)^{-1}) \le 1/\epsilon,$$

by our assumption. Note that, for any $X \in \mathbb{R}^{d_x}$,

$$\|\tilde{\eta}(X)\|_2 = \left\| A(X)^T (A(X)A(X)^T)^{-1} (h_{H+1:1}(X) - h^*(X)) \right\|_2$$
$$\le \left\| A(X)^T (A(X)A(X)^T)^{-1} \right\|_{\mathrm{op}} \|h_{H+1:1}(X) - h^*(X)\|_2$$
$$\le \left\| A(X)^T (A(X)A(X)^T)^{-1} \right\|_{\mathrm{op}} \|h_{H+1:1} - h^*\|_{\mathrm{nl}} \|X\|_2.$$

Since this holds for any $X$, we have

$$\|\tilde{\eta}\|_{\mathrm{nl}} \le \left\| A(X)^T (A(X)A(X)^T)^{-1} \right\|_{\mathrm{op}} \|h_{H+1:1} - h^*\|_{\mathrm{nl}} \le \frac{\|h_{H+1:1} - h^*\|_{\mathrm{nl}}}{\epsilon},$$

which ensures that $\tilde{\eta} \in \mathcal{F}_1$. Finally,

$$\|D_{h_1}[L(h)]\|_{\mathrm{op}} \ge \frac{\langle D_{h_1}[L(h)], \tilde{\eta}\rangle}{\|\tilde{\eta}\|_{\mathrm{nl}}} \ge \frac{\epsilon \mathbb{E}\left[\|h_{H+1:1}(X) - h^*(X)\|_2^2\right]}{\|h_{H+1:1} - h^*\|_{\mathrm{nl}}} = \frac{\epsilon(L(h) - L^*)}{\|h_{H+1:1} - h^*\|_{\mathrm{nl}}},$$

which yields

$$\|D_{h_1}[L(h)]\|_{\mathrm{op}} \|h_{H+1:1} - h^*\|_{\mathrm{nl}} \ge \epsilon(L(h) - L^*).$$

From this we can see that if we have a critical point of $L(h)$, then $\|D_{h_1}[L(h)]\|_{\mathrm{op}} = 0$ implies $L(h) = L^*$, which means that the critical point is a global minimum of $L(h)$.

## A.4   Proof of Theorem 4.2

Recall that by assumption we have $j \in \{1, \ldots, H+1\}$ such that $d_x = d_{j-1}$ and $d_y \le d_j$. Consider $D_{h_j}[L(h)]$, then for any $\eta \in \mathcal{F}_j$,

$$\langle D_{h_j}[L(h)], \eta\rangle = \mathbb{E}\left[(h_{H+1:1}(X) - h^*(X))^T J[h_{H+1:j+1}](h_{j:1}(X))\eta(h_{j-1:1}(X))\right].$$

As done in the previous theorem, for any $w \in \mathbb{R}^{d_{j-1}}$, let $A(w) = J[h_{H+1:j+1}](h_j(w))$. Since $A(w)$ has full row rank by assumption, $A(w)A(w)^T$ is invertible. Then define

$$\tilde{\eta}(w) = A(w)^T (A(w)A(w)^T)^{-1} (h_{H+1:1} - h^*) \circ h_{j-1:1}^{-1}(w),$$

so that

$$\langle D_{h_j}[L(h)], \tilde{\eta}\rangle = \mathbb{E}\left[\|h_{H+1:1}(X) - h^*(X)\|_2^2\right].$$

We need to check if $\tilde{\eta} \in \mathcal{F}_j$. It is easily checked that $\tilde{\eta}(0) = 0$. Since $J[h_{H+1:j+1}]$ is differentiable by assumption and $h_j \in \mathcal{F}_j$, $A(w)$ is differentiable, and so are $A(w)^T$ and $(A(w)A(w)^T)^{-1}$. The inverse function of a differentiable and invertible function is also differentiable, so $(h_{H+1:1} - h^*) \circ h_{j-1:1}^{-1}$ is differentiable. Hence, we can conclude that $\tilde{\eta}$ is differentiable.

As seen in the previous section,

$$\left\| A(w)^T (A(w)A(w)^T)^{-1} \right\|_{\mathrm{op}} = \sigma_{\max}(A(w)^T (A(w)A(w)^T)^{-1}) \le 1/\epsilon_2.$$

By the assumption that $h_{j-1:1}$ is invertible and $\|h_{j-1:1}(u)\|_2 \ge \epsilon_1 \|u\|_2$,

$$\|v\|_2 \ge \epsilon_1 \left\| h_{j-1:1}^{-1}(v) \right\|_2,$$

for all $v \in \mathbb{R}^{d_{j-1}}$. From this, we can see that $\left\|h_{j-1:1}^{-1}\right\|_{\mathrm{nl}} \leq 1/\epsilon_1$. For any $w \in \mathbb{R}^{d_{j-1}}$,

$$
\begin{aligned}
\|\tilde{\eta}(w)\|_2 &= \left\|A(w)^T(A(w)A(w)^T)^{-1}(h_{H+1:1} - h^*) \circ h_{j-1:1}^{-1}(w)\right\|_2 \\
&\leq \left\|A(w)^T(A(w)A(w)^T)^{-1}\right\|_{\mathrm{op}} \left\|(h_{H+1:1} - h^*) \circ h_{j-1:1}^{-1}(w)\right\|_2 \\
&\leq \left\|A(w)^T(A(w)A(w)^T)^{-1}\right\|_{\mathrm{op}} \left\|h_{H+1:1} - h^*\right\|_{\mathrm{nl}} \left\|h_{j-1:1}^{-1}(w)\right\|_2 \\
&\leq \left\|A(w)^T(A(w)A(w)^T)^{-1}\right\|_{\mathrm{op}} \left\|h_{H+1:1} - h^*\right\|_{\mathrm{nl}} \left\|h_{j-1:1}^{-1}\right\|_{\mathrm{nl}} \|w\|_2 .
\end{aligned}
$$

From this, we have

$$
\|\tilde{\eta}\|_{\mathrm{nl}} \leq \left\|A(w)^T(A(w)A(w)^T)^{-1}\right\|_{\mathrm{op}} \|h_{H+1:1} - h^*\|_{\mathrm{nl}} \left\|h_{j-1:1}^{-1}\right\|_{\mathrm{nl}} \leq \frac{\|h_{H+1:1} - h^*\|_{\mathrm{nl}}}{\epsilon_1 \epsilon_2}.
$$

Finally,

$$
\left\|D_{h_j}[L(h)]\right\|_{\mathrm{op}} \geq \frac{\left\langle D_{h_j}[L(h)], \tilde{\eta}\right\rangle}{\|\tilde{\eta}\|_{\mathrm{nl}}} \geq \frac{\epsilon_1 \epsilon_2 \mathbb{E}\left[\|h_{H+1:1}(X) - h^*(X)\|_2^2\right]}{\|h_{H+1:1} - h^*\|_{\mathrm{nl}}} = \frac{\epsilon_1 \epsilon_2 (L(h) - L^*)}{\|h_{H+1:1} - h^*\|_{\mathrm{nl}}},
$$

which yields

$$
\left\|D_{h_j}[L(h)]\right\|_{\mathrm{op}} \|h_{H+1:1} - h^*\|_{\mathrm{nl}} \geq \epsilon_1 \epsilon_2 (L(h) - L^*).
$$

## A.5 Proof of Proposition 4.3

(Theorem 4.1) By assumption, we have $d_1 \geq d_y$. Set $h_1(x) = (h^*(x), 0, \ldots, 0)$ where for every $x \in \mathbb{R}^{d_x}$, the first $d_y$ components of $h_1(x)$ are identical to $h^*(x)$, and all other components are zero. For the rest of $h_i$'s, define $h_i : \mathbb{R}^{d_{i-1}} \to \mathbb{R}^{d_i}$ to be

$$
h_i(w) = \begin{cases} (w_1, \ldots, w_{d_i}), & \text{if } d_i \leq d_{i-1}, \\ (w_1, \ldots, w_{d_{i-1}}, 0, \ldots, 0), & \text{if } d_i > d_{i-1}, \end{cases} \tag{7}
$$

for all $w \in \mathbb{R}^{d_{i-1}}$. Since $d_i \geq d_y$ for all $i$, we can check that $h_{H+1} \circ \cdots \circ h_1 = h^*$, and $h_i \in \mathcal{F}_i$ for all $i$. Moreover, for all $z \in \mathbb{R}^{d_1}$, $J[h_{H+1:2}](z)$ is all 0 except 1's in diagonal entries, so $\sigma_{\min}(J[h_{H+1:2}](z)) \geq 1$ and $h_{H+1:2}(z)$ is twice-differentiable.

(Theorem 4.2) It is given that we have $j \in \{1, \ldots, H+1\}$ such that $d_x = d_{j-1}$ and $d_y \leq d_j$. Set $h_j(x) = (h^*(x), 0, \ldots, 0)$, where the first $d_y$ components are $h^*(x)$ and the rest are zero. All the rest of $h_i$ are set as in (7). Then, it can be easily checked that $h_i \in \mathcal{F}_i$ for all $i$ and all the conditions of the theorem are satisfied.

## B Deferred Lemma

**Lemma B.1.** *Suppose we are given a data matrix $X \in \mathbb{R}^{d_x \times m}$ and an output matrix $Y \in \mathbb{R}^{d_y \times m}$, where $d_x < d_y$. Assume $XX^T$ and $YX^T$ have full ranks. Consider minimizing the empirical squared error risk:*

$$
L(W_1, \ldots, W_{H+1}) := \frac{1}{2} \|W_{H+1}W_H \cdots W_1 X - Y\|_{\mathrm{F}}^2,
$$

*where $W_k \in \mathbb{R}^{d_k \times d_{k-1}}$, $k = 1, \ldots, H+1$ are weight matrices of the linear neural network, and $d_0 = d_x$ and $d_{H+1} = d_y$ for simplicity in notation. Also let $W$ denote the tuple $(W_1, \ldots, W_{H+1})$. Then, any critical point of $L(W)$ that is not a local minimum is a saddle point.*

*Proof.* For this lemma, we separate the proof into two cases: $W_H \cdots W_1 \neq 0$ and $W_H \cdots W_1 = 0$. The crux of the proof is to show that any critical point cannot be a local maximum. Then, any critical point is either a local minimum or a saddle point, so the conclusion of this lemma follows.

In case of $W_H \cdots W_1 \neq 0$, we use some of the results in Kawaguchi (2016) and examine the Hessian of $L(W)$ with respect to $\mathrm{vec}(W_{H+1}^T)$, where $\mathrm{vec}(A)$ denotes vectorization of matrix $A$.

Let $D_{\text{vec}(W_{H+1}^T)}L(W)$ be the partial derivative of $L(W)$ with respect to $\text{vec}(W_{H+1}^T)$ in numerator layout. It was shown by Kawaguchi (2016, Lemma 4.3) that the Hessian matrix

$$\mathcal{H}(W) = D_{\text{vec}(W_{H+1}^T)}\left(D_{\text{vec}(W_{H+1}^T)}L(W)\right)^T = \left(I \otimes (W_H \cdots W_1 X)(W_H \cdots W_1 X)^T\right)$$
$$= \left(I \otimes W_H \cdots W_1 X X^T W_1^T \cdots W_H^T\right),$$

where $\otimes$ denotes the Kronecker product of two matrices. Notice that $\mathcal{H}(W)$ is positive semidefinite. Since $XX^T$ is full rank, whenever $W_H \cdots W_1 \neq 0$ there exists a strictly positive eigenvalue in $\mathcal{H}(W)$, which means that there exists an increasing direction. So $W$ cannot be a local maximum.

The case where $W_H \cdots W_1 = 0$ requires a bit more careful treatment. Note that this case corresponds to where we have degenerate critical points, which are in many cases much harder to handle.

For any arbitrary $\epsilon > 0$, we describe a procedure that perturbs the matrices $W_1, \ldots, W_{H+1}$ by perturbations sampled from Frobenius norm balls of radius $\epsilon$ centered at 0, which we will denote as $\mathcal{B}_i(\epsilon), i = 1, \ldots, H+1$. Let $\mathcal{U}(\mathcal{B}_i(\epsilon))$ be the uniform distribution over the ball $\mathcal{B}_i(\epsilon)$. The algorithm goes as the following:

1. For $i \in \{1, \ldots, H+1\}$

   1.1. Sample $\Delta_i \sim \mathcal{U}(\mathcal{B}_i(\epsilon))$, and define $V_i = W_i + \Delta_i$.
   1.2. If $W_{H+1} \cdots W_{i+1} V_i \cdots V_1 \neq 0$, stop and return $i^* = i$.

First, recall that the set of rank-deficient matrices have Lebesgue measure zero, so for any sample $\Delta_i \sim \mathcal{U}(\mathcal{B}_i(\epsilon))$, $V_i = W_i + \Delta_i$ has full rank with probability 1. If we proceed the for loop until $i = H+1$, we have a full-rank $V_{H+1} \cdots V_1$ with probability 1, which means that the algorithm must return $i^* \in \{1, \ldots, H+1\}$ with probability 1. Notice that before and after the $i^*$-th iteration, we have

$$W_{H+1} \cdots W_{i^*} V_{i^*-1} \cdots V_1 = 0,$$
$$W_{H+1} \cdots W_{i^*+1} V_{i^*} \cdots V_1 = W_{H+1} \cdots W_{i^*+1}(W_{i^*} + \Delta_{i^*})V_{i^*-1} \cdots V_1 \neq 0.$$

This means that if we define $\hat{\Delta} = W_{H+1} \cdots W_{i^*+1}\Delta_{i^*}V_{i^*-1} \cdots V_1$, then $\hat{\Delta} \neq 0$. Also, notice that

$$W_{H+1} \cdots W_{i^*+1}(W_{i^*} - \Delta_{i^*})V_{i^*-1} \cdots V_1 = -\hat{\Delta}.$$

Now, define two points

$$U^{(1)} = (V_1, \ldots, V_{i^*-1}, W_{i^*} + \Delta_{i^*}, W_{i^*+1}, \ldots, W_{H+1}),$$
$$U^{(2)} = (V_1, \ldots, V_{i^*-1}, W_{i^*} - \Delta_{i^*}, W_{i^*+1}, \ldots, W_{H+1}),$$

and notice that they are all in the neighborhood of $W$, that is, the Cartesian product of $\epsilon$-radius balls centered at $W_1, \ldots, W_{H+1}$. Moreover, we have

$$L(W) = \frac{1}{2}\|0 \cdot X - Y\|_F^2 = \frac{1}{2}\|Y\|_F^2,$$
$$L(U^{(1)}) = \frac{1}{2}\left\|\hat{\Delta}X - Y\right\|_F^2 = \frac{1}{2}\|Y\|_F^2 + \frac{1}{2}\left\|\hat{\Delta}X\right\|_F^2 - \left\langle\hat{\Delta}X, Y\right\rangle,$$
$$L(U^{(2)}) = \frac{1}{2}\left\|-\hat{\Delta}X - Y\right\|_F^2 = \frac{1}{2}\|Y\|_F^2 + \frac{1}{2}\left\|\hat{\Delta}X\right\|_F^2 + \left\langle\hat{\Delta}X, Y\right\rangle,$$

from which we can see that at least one of $L(W) < L(U^{(1)})$ or $L(W) < L(U^{(2)})$ must hold. This shows that for any $\epsilon > 0$, there is a point $U$ in $\epsilon$-neighborhood of $W$ with a strictly greater function value $L(U)$. This proves that $W$ cannot be a local maximum. $\square$

## C    DEFERRED PROOFS

### C.1    PROOF OF PROPOSITION 3.1

In case of $k < \min\{d_x, d_y\}$, we can decompose the loss function in the following way:

$$
\begin{aligned}
\|RX - Y\|_{\mathrm{F}}^2 &= \left\|RX - YX^T(XX^T)^{-1}X + YX^T(XX^T)^{-1}X - Y\right\|_{\mathrm{F}}^2 \\
&= \left\|RX - YX^T(XX^T)^{-1}X\right\|_{\mathrm{F}}^2 + \left\|YX^T(XX^T)^{-1}X - Y\right\|_{\mathrm{F}}^2 \\
&\quad + 2\operatorname{tr}((YX^T(XX^T)^{-1}X - Y)(RX - YX^T(XX^T)^{-1}X)^T).
\end{aligned}
$$

Let us take a close look into the last term in the RHS. Note that $YX^T(XX^T)^{-1}X$ is the orthogonal projection of $Y$ onto $\operatorname{row}(X)$, so each row of $YX^T(XX^T)^{-1}X - Y$ must be in $\operatorname{null}(X)$. Also,

$$
(RX - YX^T(XX^T)^{-1}X)^T = X^T(R^T - (XX^T)^{-1}XY^T).
$$

It is $X^T$ right-multiplied with some matrix, so its columns must lie in $\operatorname{col}(X^T) = \operatorname{row}(X)$. By the fact that $\operatorname{null}(X)^\perp = \operatorname{row}(X)$,

$$
(YX^T(XX^T)^{-1}X - Y)(RX - YX^T(XX^T)^{-1}X)^T = 0,
$$

thus

$$
L_0(R) = \frac{1}{2}\left\|RX - YX^T(XX^T)^{-1}X\right\|_{\mathrm{F}}^2 + \frac{1}{2}\left\|YX^T(XX^T)^{-1}X - Y\right\|_{\mathrm{F}}^2
$$

holds.

Now, (2) becomes a problem of minimizing $\left\|RX - YX^T(XX^T)^{-1}X\right\|_{\mathrm{F}}^2$ subject to the rank constraint $\operatorname{rank}(R) \le k$. The optimal solution for this is obtained when $RX$ is the $k$-rank approximation of $YX^T(XX^T)^{-1}X$. Then, $k$-rank approximation of $YX^T(XX^T)^{-1}X$ can be expressed as $\hat{U}\hat{U}^T YX^T(XX^T)^{-1}X$, where $\hat{U}$ is unique due to our assumption that all singular values are distinct. Therefore,

$$
R^* = \hat{U}\hat{U}^T YX^T(XX^T)^{-1}
$$

is the unique global minimum solution of (2) when $k < \min\{d_x, d_y\}$.

### C.2    PROOF OF LEMMA 3.2

$$
\begin{aligned}
&L(W_1, \ldots, W_{i-1}, W_i + \Delta_i, W_{i+1}, \ldots, W_{H+1}) \\
={}& \frac{1}{2}\left\|W_{H+1}\cdots W_{i+1}(W_i + \Delta_i)W_{i-1}\cdots W_1 X - Y\right\|_{\mathrm{F}}^2 \\
={}& \frac{1}{2}\left\|W_{H+1}\cdots W_1 X - Y + W_{H+1}\cdots W_{i+1}\Delta_i W_{i-1}\cdots W_1 X\right\|_{\mathrm{F}}^2 \\
={}& L(W) + \operatorname{tr}((W_{H+1}\cdots W_{i+1}\Delta_i W_{i-1}\cdots W_1 X)^T(W_{H+1}\cdots W_1 X - Y)) + O(\|\Delta_i\|_{\mathrm{F}}^2) \\
={}& L(W) + \operatorname{tr}(W_{i+1}^T\cdots W_{H+1}^T(W_{H+1}\cdots W_1 X - Y)X^T W_1^T\cdots W_{i-1}^T\Delta_i^T) + O(\|\Delta_i\|_{\mathrm{F}}^2).
\end{aligned}
$$

From this, we can conclude that

$$
\frac{\partial L}{\partial W_i} = W_{i+1}^T\cdots W_{H+1}^T(W_{H+1}\cdots W_1 X - Y)X^T W_1^T\cdots W_{i-1}^T.
$$

### C.3    PROOF OF LEMMA 3.3

1. Since $A^T A \succeq \sigma_{\min}^2(A)I$, $B^T A^T AB \succeq \sigma_{\min}^2(A)B^T B$. Then

$$
\|AB\|_{\mathrm{F}}^2 = \operatorname{tr}(B^T A^T AB) \ge \sigma_{\min}^2(A)\operatorname{tr}(B^T B) = \sigma_{\min}^2(A)\|B\|_{\mathrm{F}}^2.
$$

2. Since $BB^T \succeq \sigma_{\min}^2(B)I$, $ABB^T A^T \succeq \sigma_{\min}^2(B)AA^T$. Then

$$
\|AB\|_{\mathrm{F}}^2 = \operatorname{tr}(B^T A^T AB) = \operatorname{tr}(ABB^T A^T) \ge \sigma_{\min}^2(B)\operatorname{tr}(AA^T) = \sigma_{\min}^2(B)\|A\|_{\mathrm{F}}^2.
$$

