# OpenReview forum: "Global Optimality Conditions for Deep Neural Networks"
_ICLR.cc/2018/Conference — Accept (Poster)_

### Official Review · AnonReviewer1 · 2017-11-25
**A nice but incremental theoretical paper which contributes to better understand the optimization of deep network**

**Rating:** 7
**Confidence:** 4

**Review:**

Summary:
The paper gives theoretical results regarding the existence of local minima in the objective function of deep neural networks. In particular:
- in the case of deep linear networks, they characterize whether a critical point is a global optimum or a saddle point by a simple criterion. This improves over recent work by Kawaguchi who showed that each critical point is either a global minimum or a saddle point (i.e., none is a local minimum), by relaxing some hypotheses and adding a simple criterion to know in which case we are.
- in the case of nonlinear network, they provide a sufficient condition for a solution to be a global optimum, using a function space approach.

Quality:
The quality is very good. The paper is technically correct and nontrivial. All proofs are provided and easy to follow.

Clarity:
The paper is very clear. Related work is clearly cited, and the novelty of the paper well explained. The technical proofs of the paper are in appendices, making the main text very smooth.

Originality:
The originality is weak. It extends a series of recent papers correctly cited. There is some originality in the proof which differs from recent related papers.

Significance:
The result is not completely surprising, but it is significant given the lack of theory and understanding of deep learning. Although the model is not really relevant for deep networks used in practice, the main result closes a question about characterization of critical points in simplified models if neural network, which is certainly interesting for many people.

---

> ### Author Response · Authors · 2018-01-05
> **Response to the review**
>
> Thank you very much for the review. We especially appreciate that the reviewer recognized the quality and clarity of our paper. Since we think that the reviewer has a good understanding of our paper and the reviewer did not have any specific questions for us, we would like to comment a little more about the significance of this paper.
>
> We would like to emphasize that our results extend the previous “existence” theorems in Kawaguchi’16 to “computational” theorems that can actually help optimization of linear neural networks. In other words, previous works on linear neural networks only proved that there exist only global minima and saddle points, whereas we provide *computable* tests for distinguishing global minima from others. This means that we can use the conditions while running optimization algorithms to determine which kind of critical point we are at, and choose the next action accordingly.
>
> Aside from this computational perspective, considering that optimizing deep linear networks is a nonconvex problem, our checkable global optimality conditions are interesting in their own right, because in the worst cases even checking local optimality of nonconvex problems could be intractable.

---

### Official Review · AnonReviewer2 · 2017-11-26
**This paper studies some theoretical properties of deep linear networks. Compared to state of the art this paper has fewer assumptions and much shorter and more concise proofs. This paper does build on a few previous results but still in my view has nice contributions. In summary, this is a short, neat and concise paper, making it an ideal conference submission.**

**Rating:** 8
**Confidence:** 5

**Review:**


-I think title is misleading, as the more concise results in this paper is about linear networks I recommend adding linear in the title i.e. changing the title to … deep LINEAR networks

- Theorems 2.1, 2.2 and the observation (2) are nice!

- Theorem 2.2 there is no discussion about the nature of the saddle point is it strict? Does this theorem imply that the global optima can be reached from a random initialization? Regardless of if this theorem can deal with these issues, a discussion of the computational implications of this theorem is necessary.

- I’m a bit puzzled by Theorems 4.1 and 4.2 and why they are useful. Since these results do not seem to have any computational implications about training the neural nets what insights do we gain about the problem by knowing this result? Further discussion would be helpful.

---

> ### Author Response · Authors · 2018-01-05
> **Response to the review**
>
> We thank the reviewer for their effort in reviewing our paper and for the encouragement.
>
> We agree that the main content of this paper is about linear networks, but since we also have some preliminary results on nonlinear case (albeit in the abstract functional space setting), we kept a more general title to serve as a small indicator of this.
>
> Our theorems do not imply anything about strict saddle property of linear neural networks. In fact, it was shown by Kawaguchi’16 that in linear neural networks there are many non-strict saddle points i.e. saddle points without negative eigenvalues. So, our theorems do not imply that global optima can always be reached by random initialization and just running SGD-like methods. However, there is actually some computational implication of these theorems; with these global optimality conditions, whenever we reach a critical point we can always efficiently check if it's a global minimum or a saddle point. If we are indeed at a global minimum, we can just return the current point and terminate. If we are at a saddle, we can then intentionally add random perturbations to the point and try to escape the saddle.
>
> Our nonlinear results are in a function space setting, so their implications are limited in computational aspects. However, we believe that these results can be good initial steps from the theoretical point of view; for example, we can see that one of the sufficient conditions for global optimality is the Jacobian matrix being full rank. Given that a nonlinear function can locally be linearly approximated using Jacobians, this connection is already interesting. An extension of the function space viewpoint to cover different architectures or design new architectures (that have “better” properties when viewed via the function space view) should also be possible and worth studying.

---

### Official Review · AnonReviewer3 · 2017-11-30

**Rating:** 5
**Confidence:** 5

**Review:**

The paper gives sufficient and necessary conditions for the global optimality of the loss function of deep linear neural networks. The paper is an extension of Kawaguchi'16. It also provides some sufficient conditions for the non-linear cases.

I think the main technical concerns with the paper is that the technique only applies to a linear model, and it doesn't sound the techniques are much beyond Kawaguchi'16. I am happy to see more papers on linear models, but I would expect there are more conceptual or technical ingredients in it. As far as I can see, the same technique here will fail for non-linear models for the same reason as Kawaguchi's technique. Also, I think a more interesting question might be turning the landscape results into an algorithmic result --- have an algorithm that can guarantee to converge a global minimum. This won't be trivial because the deep linear networks do have a lot of very flat saddle points and therefore it's unclear whether one can avoid those saddle points.

---

> ### Author Response · Authors · 2018-01-05
> **Response to the review**
>
> We appreciate your efforts for reviewing our paper. We admit that the key results of our paper are for the linear case, and global optimality conditions do not directly apply to nonlinear models that are used in practice.
>
> But we believe that there is strong value in trying to fully understand the linear case, as this offers building blocks towards investigating nonlinear models, for instance in helping identify structures and settings that help us in quest for understanding realistic architectures.
>
> We note that our results extend previous papers such as Kawaguchi’16 and Hardt and Ma’17 in a substantial manner: our results have direct computational implications and provide a “complete” picture of the landscape of optimal for the deep linear case.
>
> More concretely, previous works on this topic only show that there are only global minima or saddle points; these results are “existence” results and there is little computational gain one can get from them. In contrast, we present *efficiently checkable* conditions for distinguishing the two different types of critical points (global min or saddle): one can even use these conditions while running optimization algorithms to check whether the critical points we encounter are saddle points or not, if desired.
>
> More broadly, we would like to emphasize again that since deep linear networks is itself a nonconvex problem, having a checkable necessary and sufficient global optimality condition is quite interesting, because in general for nonconvex problems, not only global but merely verifying even local optimality can be computationally intractable.
>
> Developing a provably convergent algorithm based on our results is a very good research direction to improve our understanding of the loss surface. Although our paper does not answer these questions, we appreciate the reviewer’s advice for this valuable future research direction.

---

### Decision · Program_Chairs · 2018-01-29
**ICLR 2018 Conference Acceptance Decision**

**Decision:**

Accept (Poster)

**Comment:**

Understanding global optimality conditions for deep nets even in the restricted case of linear layers is a valuable contribution. Please add clarifications to ways in which the paper goes beyond the results of Kawaguchi'16, which was the main concern expressed by the reviewers.